

# An adaptive scale Gaussian filter to explain White's illusion from the viewpoint of lightness assimilation for a large range of variation in spatial frequency of the grating and aspect ratio of the targets

Soma Mitra[1], Debasis Mazumdar[1], Kuntal Ghosh[2] and Kamales Bhaumik[1]

[1] Center for Development of Advanced Computing, Kolkata, India
[2] Indian Statistical Institute, Kolkata, India

## ABSTRACT

The variation between the actual and perceived lightness of a stimulus has strong dependency on its background, a phenomena commonly known as lightness induction in the literature of visual neuroscience and psychology. For instance, a gray patch may perceptually appear to be darker in a background while it looks brighter when the background is reversed. In the literature it is further reported that such variation can take place in two possible ways. In case of stimulus like the Simultaneous Brightness Contrast (SBC), the apparent lightness changes in the direction opposite to that of the background lightness, a phenomenon often referred to as lightness contrast, while in the others like neon colour spreading or checkerboard illusion it occurs opposite to that, and known as lightness assimilation. The White's illusion is a typical one which according to many, does not completely conform to any of these two processes. This paper presents the result of quantification of the perceptual strength of the White's illusion as a function of the width of the background square grating as well as the length of the gray patch. A linear filter model is further proposed to simulate the possible neurophysiological phenomena responsible for this particular visual experience. The model assumes that for the White's illusion, where the edges are strong and quite a few, i.e., the spectrum is rich in high frequency components, the inhibitory surround in the classical Difference-of-Gaussians (DoG) filter gets suppressed, and the filter essentially reduces to an adaptive scale Gaussian kernel that brings about lightness assimilation. The linear filter model with a Gaussian kernel is used to simulate the White's illusion phenomena with wide variation of spatial frequency of the background grating as well as the length of the gray patch. The appropriateness of the model is presented through simulation results, which are highly tuned to the present as well as earlier psychometric results.

Corresponding author
Kuntal Ghosh, kuntal@isical.ac.in

## INTRODUCTION

Studies on visual illusions generally help in the formulation and testing of empirical models on visual perception. Plausible neural circuit, on the basis of direct or indirect evidences from neurophysiology, is often cited as a support to the empirical model. For some simple visual illusions, the visual presentations consist of a Background (uniform field across the display) and the Target Patches at specified colour values. A common example of such an illusion is the "Simultaneous Brightness Contrast", in which two targets with identical gray values are placed in different backgrounds. Target in the dark background appears lighter than the target in the white background. This is nicely explained with the empirical model of lateral inhibition (*Kuffler, 1953*). Here one assumes that the receptive field of, say a ganglionic cell, spread over the primary receptors in the retina, may be divided in a central and a peripheral region. For the same stimulus across the receptive field, the central and the peripheral regions send signals of opposite nature to the particular ganglion cell. If the former sends the excitatory signal, the later sends the inhibitory signal or vice versa. The model received wide attention because the results obtained from experimental neurophysiology seemed to provide a support to it (*Ratliff & Hartline, 1959*). The model was subsequently refined as "Difference of Gaussian" or DOG model (*Rodieck & Stone, 1965*) and also as "Laplacian of Gaussian" or LOG model (*Marr, 1982*). The lateral inhibitory process have already been used in developing model for various visual phenomena associated with lightness illusions (*Macknik, Martinez-Conde & Haglund, 2000*; *Troncoso, Macknik & Martinez-Conde, 2005*; *Troncoso et al., 2007*; *Troncoso, Macknik & Martinez-Conde, 2009*)

There are a number of illusions for which the visual presentations contain another component, known as "Inducing Grating". Some of these, like the sinusoidal grating induction (*McCourt, 1982*), are explained with the help of lateral inhibition. However, there are others, like the square grating background in White's effect, which are not explainable with "lateral Inhibition" or DOG models. One such example is shown in Fig. 1, which has been adapted from *De Weert & Spillmann (1995)*. Though this illusion was designed to show the asymmetry between induced lightness and induced darkness, it also beautifully illustrates the phenomenon of assimilation. Arrowhead 1 in the figure shows a target in the dark background, while arrowhead 2 shows a target (with same gray value as the previous one) in the white background. In this illusion the target at arrow 1 appears darker than the target at arrow 2, totally violating the principle of lateral inhibition. An alternative empirical model, known as "lightness assimilation", may be proposed in such cases. The term "lightness" needs to be defined. Appearance of an object depends not only on the luminance (luminous intensity over a given area and direction), but also on the reflectance of the object. Brightness is defined as "apparent luminance", while lightness is known as the "apparent reflectance". Brightness ranges from "dim" to "bright", while lightness ranges from "dark" to "light". So long the visible illumination is uniform, lightness follows brightness. The two properties are different when the scenario is non-uniformly illuminated. To explain the occurrence of illusions of the type, given in Fig. 1, one has to assume that the targets assimilate lightness from the surrounding or in other words the

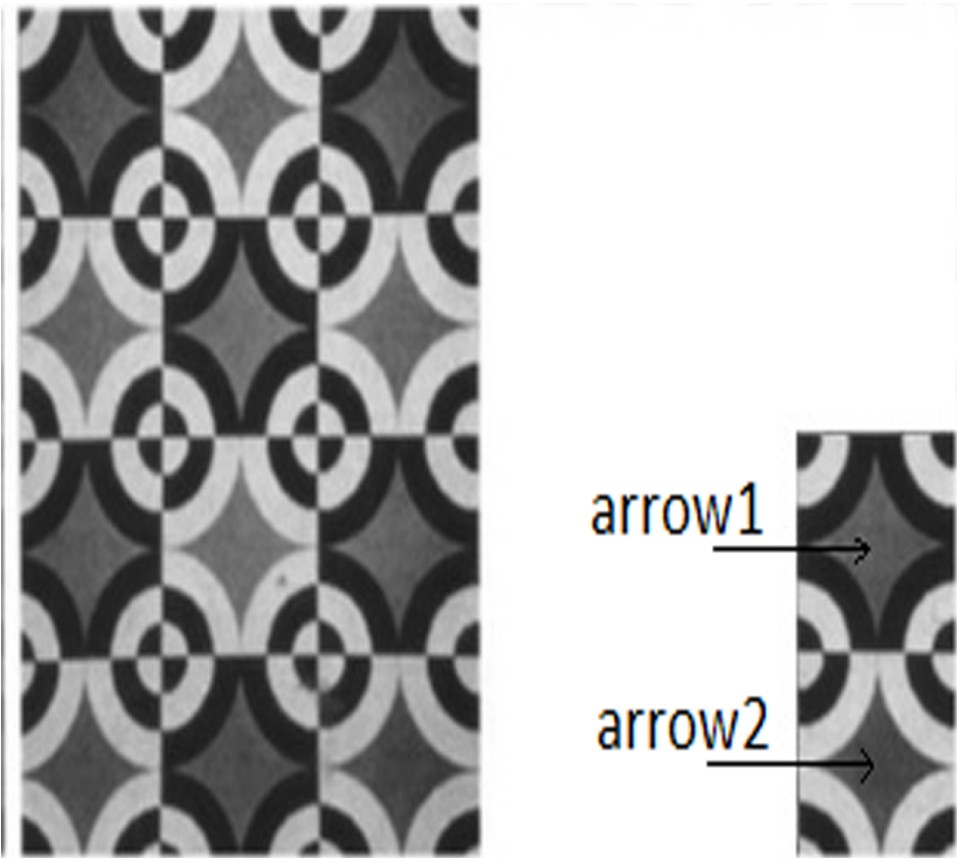

**Figure 1   Example of an illusion, exhibiting lightness assimilation.** For details, see text.

lightness at any point on the target patch is obtained by averaging the lightness over its surrounding. Such an averaging, also known as smoothing, is generally simulated through a Gaussian filter. Hence there is a popular belief that the visual presentations are perceived by the human brain after being modulated either through a DOG filter (contrast) or through a Gaussian filter (assimilation) or through an weighted sum of both (*Young, 1987*). It is also possible that in reality the human visual perception may not follow either of these alternatives. However, there is a relentless controversy (*Kingdom, 2011*) between these two processes and it will continue until the exact neural correlates of the visual perception are experimentally well established.

The present paper contains the report of our experimental and theoretical studies on the well-known "White's illusion" (*White, 1979*; *White, 1981*), as shown in Fig. 2. Here the inducing grating consists of alternate black and white bars of equal width, characterized by its spatial frequency (which is generally expressed in cpd or cycles of repetitions per degree visual angle). Two target patches of equal length and identical gray values are taken. Both are placed co-axially, one along a white bar and the other along a dark bar. It is observed that the target which is mostly surrounded by black bars looks darker than the target which is mostly surrounded by the white bar. Since the observations cannot be explained through
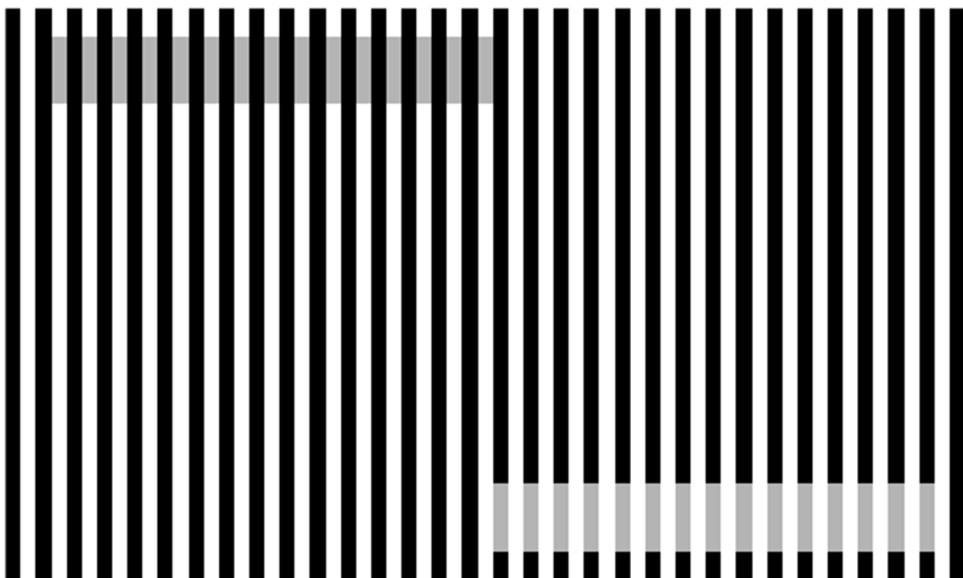

**Figure 2  Example of White's illusion.** For details, see text.

the model of "lateral inhibition", a model of "assimilation" is invoked for the explanation. The model of assimilation provides support for the perception of White's illusion over an wide range. Our experimental data show that the standard deviation (also known as the scale factor) of the Gaussian filter remains constant for a particular value of the spatial frequency of the inducing grating for an wide range of length of the test patch. The question is whether the model of assimilation is adequate for all ranges of length and width of the target patch. For example, if the length of the target becomes comparable to (or smaller than) its width, would it be possible to account for the magnitude of White's illusion with a single Gaussian filter? The present paper shows that the answer is affirmative, but, of course, with a price. It is observed that once the length of the patch is comparable to its width, the scale factor is to be adapted to a higher value. Hence we have used an adaptive scale Gaussian filter to fit the data on White's illusion from the viewpoint of lightness assimilation for a large range of variation in spatial frequency of the grating and aspect ratio of the targets.

The organization of the paper is as follows. In the beginning, we present the results of psychometric tests in our laboratory, following the methodology of two earlier papers (*Shi et al., 2013*; *Troncoso, Macknik & Martinez-Conde, 2005*). We performed two different experiments. In the first experiment, we measured the effect of the illusion for five different spatial frequencies of the inducing grating, while keeping the length of the target as fixed. In the second experiment, we had varied the length of the target to four different values at two fixed spatial frequencies. Purpose of the first experiment is to estimate the illusory component of perception for various spatial frequencies. Motivation of the second experiment comes from our emphasis in establishing assimilation as a dominant feature of White's illusion. The section, entitled 'Materials and Methods', gives the experimental set

up in detail. The section on 'Results' provides the data obtained from our experiments and comparison of those with existing data. Subsequent section on 'Proposed Model' describes how we have tried to explain the results with an adaptive scale Gaussian filter model. In the section on 'Discussion', we provide justification in proposing a new model in spite of the fact that there are good models from reputed groups working in this field. Finally there is a section on 'Conclusion'.

## MATERIALS AND METHODS

We report here two sets of data, obtained from our experiments. Two relevant parameters for our experiments are the spatial frequency of the bars and the length of the target patches. Spatial frequency is given in the unit of cpd or cycles per degree of angle subtended at the eye. Lengths and breadths of the patches may be given in pixels or in degree. In the following paragraph, we shall give the algorithm for the conversion of bar width to cpd or pixels to degree. In the first set, the spatial frequency was varied from 2.97 cpd to 0.368 cpd, whereas the target length was kept constant at 70 pixels. In the second set the target length was varied from 16 pixels to two pixels while the spatial frequency was held constant at 1.47 cpd and 0.738 cpd, respectively. In both the cases psychometric tests were conducted with six subjects including three adult males and three adult females. Four were naïve while two were chosen from among the authors. Each experimental session was of about 30 minutes' duration and five such sessions completed a full cycle of experiment. Written consents were obtained from all the subjects. The experiments have been approved by Scientific and Technical Advisory Committee (STAC), C-DAC.

### Variation of spatial frequency of the inducing grating

The experimental arrangements were designed identical to that described in *Shi et al. (2013)* and *Troncoso, Macknik & Martinez-Conde (2005)*. To stabilize the subjects' heads, a chin rest was placed at a distance $d$ from a linearized video monitor (HP Compaq LE 2002X with resolution 1,024 × 1,024 pixels). The value of $d$ was chosen as 57 cm, because it may be shown from simple trigonometry that at such a distance an image of width (or length) of one cm. subtends a visual angle of approximately 1 degree. By measuring the pixel width in centimeter and by counting number of cycles per centimeter, one may convert those to cpd. For example, for bars of width two pixels, there are 5.5 cycles within a distance of 1.85 centimeters. Similarly for bars of width 32 pixels, there are 2.5 cycles within a distance of 13.6 centimeters. Accordingly widths of two, four, eight, 12, 16 and 32 pixels would correspond to 2.97, 1.47, 0.738, 0.493, 0.368 and 0.184 cpd respectively. Subjects binocularly viewed the visual presentations, keeping their heads fixed on the chin rest. In these experiments, the subjects visually compare the brightness or lightness of two targets, namely the *standard* and the *comparator*. In our experiments, the *standard* was a target, containing a number of segments. Each segment could be distinguished from the other by its intensity of gray value. For example, our striped *standard* was divided into 11 segments of varying intensity. These values were kept fixed during the entire experiment, although the order of appearance of these 11 segments within the *standard* was scrambled pseudo-randomly. In our experiments, the *comparators* are the targets of identical gray

values, taken on the black and white bars and hence these are the generators of the White's illusion, as shown clearly in Fig. 2. It is clear from Fig. 2, that the *comparators* that appear darker are bordered by more black than white, whereas the *comparators* that appear lighter are bordered by more white than black. Though the gray values of various segments of the *standard* are kept fixed (although scrambled) during one set of experiment, the *comparators* are changed from trial to trial, either by varying its width or length. Width of a *comparator* is always equal to the width of the black and white bars of the inducing grating. Hence the variation of the width of the *comparator* is synonymous to the variation of the spatial frequency of the grating. Subjects visually compare between the *comparator* and the *standard* on an uniform background with 50% relative luminance (equivalent to a gray value of 128). At the beginning of each trial, the subject was instructed to fix attention on a central red cross (1° within a 3.5° fixation window). After a lapse of 1 second, both *comparator* and *standard* appeared on the screen simultaneously. One of them was centered at 7° to the left while the other centered at 7° to the right of the central cross. Two-alternative forced-choice (2AFC) paradigm, introduced by *Fechner (1889)*, was used in these lightness discrimination experiments. If the *comparator* appeared to be lighter than the *standard*, subjects had to press key number One, otherwise they had to press key number Two. Figure 3A shows the screen design. The comparator was designed such that the absolute gray values of the black bars, white bars and the comparators were 0, 256 and 128, respectively. Perceived lightness of the comparators were strongly influenced by the lightness of the surrounding bars. Since the widths of the coaxial bars modulate the perceived lightness of the comparators, we considered five different widths. Figure 3B shows three such stimulus presentations. For the smallest width (2.97 cpd), 11 number of bars (white + dark) could be accommodated within the stimulus, whereas for the largest width (0.368 cpd), the number of bars had to be reduced to five. This variation in the number of bars was necessary to ensure that the region of comparison always be within 7° around the central cross mark. Two red vertical indicator lines were displayed 6° from the top or from the bottom end of both the standard and the comparator, in order to confine the attention of the subjects within the specific region of interest. This is shown in Fig. 3B for three different cases. Stimuli (comparator and standard) appeared on the display for a duration of 3 seconds. Subjects need not had to wait to give their judgments till the stimuli disappeared from the display. Red lines indicate the region of interest in the comparator to be judged against the nearest segment of the standard. The random choice of the selection of the region of interest ensured unbiased and uniform probability distribution. Perceived difference of brightness between the comparator and the standard depends on the actual difference of brightness between those and also the psychophysical effect on the subject. If the actual difference in the brightness of the co-occurring comparator and the standard is zero, the apparent perceived difference is then entirely due to the psychophysical effect.

To keep the subjects unbiased, alert and attentive and also to avoid the fatigue during the experiments, various parameters were randomly changed during the display. A number of criteria were used in designing the experimental sub-session as listed below:

(a) The standard had 11 segments with absolute gray values 13, 36, 59, 82, 105, 128, 150, 173, 196, 219 and 242. Equivalently the relative luminance of these segments were 5%,

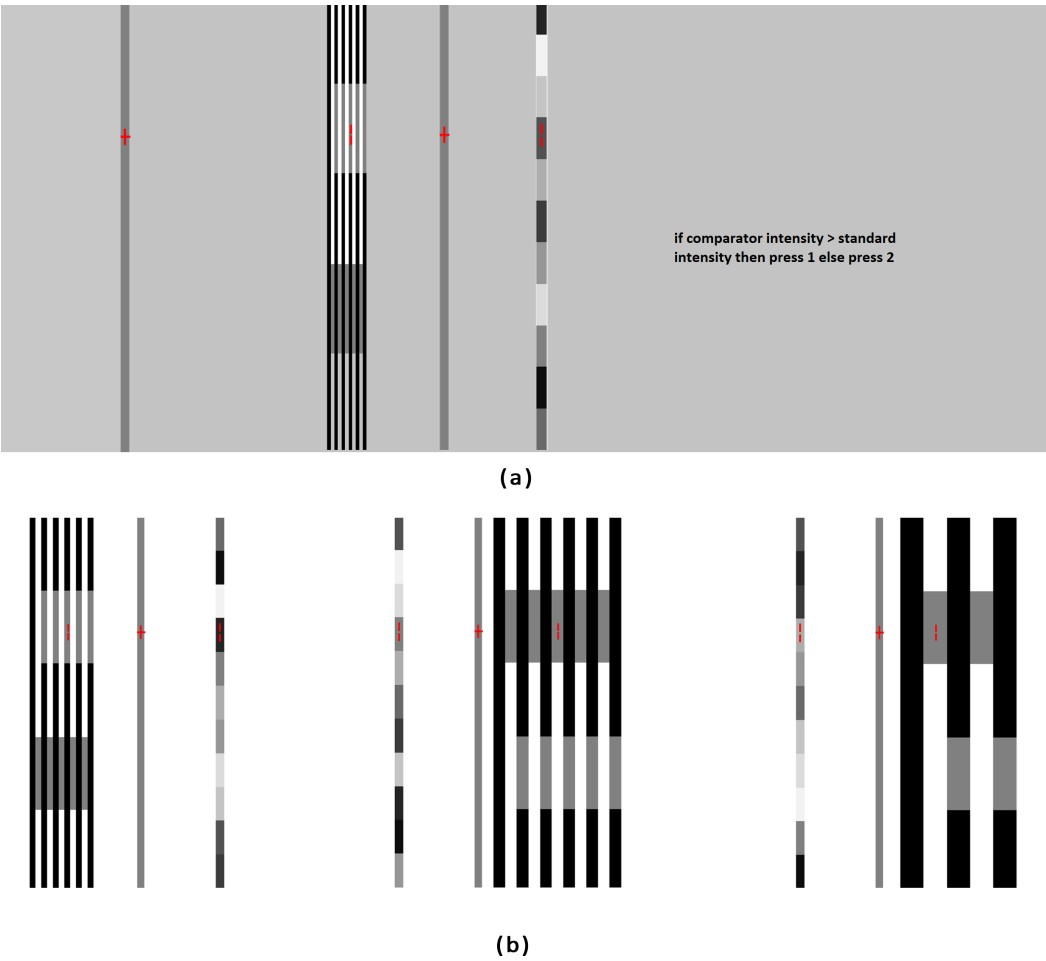

if comparator intensity > standard
intensity then press 1 else press 2

(a)

(b)

**Figure 3** **Screen design of the psychophysical experiment.** (A) Screen design of the psychophysical experiment, (B) three different stimulus presentations of the lightness discrimination experiment.

14%, 23%, 32%, 41%, 50%, 59%, 68%, 77%, 86% and 95%, respectively. (Relative luminance of 100% corresponds to absolute gray value 255). *Number of variations = 11*

(b) The subjects were exposed to a light appearing comparator (coaxial with black bar and flanked by white bars on both sides) in one half of the trials and a dark appearing comparator (coaxial with white bar) in the other half of the trials. *Number of variations = 2.*

(c) The comparator appeared half the time on the left and half the time on the right of the standard during a complete session. *Number of variations = 2.*

(d) The fixation marker was presented half the time on the top of the screen and half the time at the bottom of the screen randomly. *Number of variations = 2.*

(e) In order to have equal probability of occurrence of each variation, total number of trials in a sub-session should be $11 \times 2 \times 2 \times 2 = 88$. Instead we have taken the number of trials in a sub-session as $88 \times 2 = 176$, so that number of occurrence of each combination is increased.

One session of experiment consists of five sub-sessions. In each of the sub-session the spatial frequency of the grating was kept fixed. The frequency was varied over the range 2.97, 1.47, 0.738, 0.493 and 0.368 cpd. The length of the comparator was kept fixed at 70 pixels or 12.075 degrees. So the total number of experiments in a session is $176 \times 5 = 880$. Several such stimuli are shown in Fig. 3B.

## Variation of the length of the comparators

In a second experiment, we have tested whether the features of White's illusion are compatible with the model of "lightness assimilation" for an wide range of the length of the comparators. Over here the spatial frequency of the inducing grating is kept fixed, while the length of the comparators is gradually changed. We have repeated this experiment for two different values of spatial frequencies, namely 1.47 cpd and 0.738 cpd. For each of these frequencies, the lengths of the comparators were varied over four values, namely, 16, eight, four and two pixels (corresponding to 2.76, 1.38, 0.69 and 0.345 degrees). The procedure of the experiments was identical to the previous one. The same six subjects (three males and three females) performed the experiments under identical conditions. Each sub-session, as before, consisted of 176 trials. So the total session required $176 \times 4 = 704$ trials. The subjects gave their judgments following 2AFC protocol. Written consents were obtained from all the subjects. When the length of the comparator is very small (like four or two pixels), some of the sub-sessions were repeated by keeping the red indicator in one set, while removing the red indicator in the repeated set. No perceivable difference in the outcome of the experiment in the two sets was observed, indicating that there was no problem in the visibility of the targets in presence of the indicator.

## RESULTS

Experiments, with 2AFC protocol, determine the subjective response thresholds of the performers of the experiments, which are essentially the comparator intensities required to produce a given level of performance. Performance of the subject improves as the actual difference in intensity of the comparator and the standard increases. Purpose of these experiments is to measure two main parameters.

(a) "point of subjective equality" (PSE): the actual intensities of the comparator and the standard, when these appear to be same to the subject.

(b) Subjective ability to just discriminate between the intensities of the comparator and the standard. The former is known as "bias", while the later determines the "discrimination threshold".

## Variation with spatial frequency

Psychometric curves, given in Fig. 4A are obtained by fitting the data with logistic functions using a maximum likelihood procedure. The function FitPsycheCurveLogit (used in MATLAB) is designed to fit a basic psychometric curve using a general linear model. The function uses **glmfit** to fit a binomial distribution with a logit link function. It is basically a cumulative Gaussian. The mean and variance of the Gaussian are assigned as the subject "bias" and subjective "discrimination threshold". The function may take

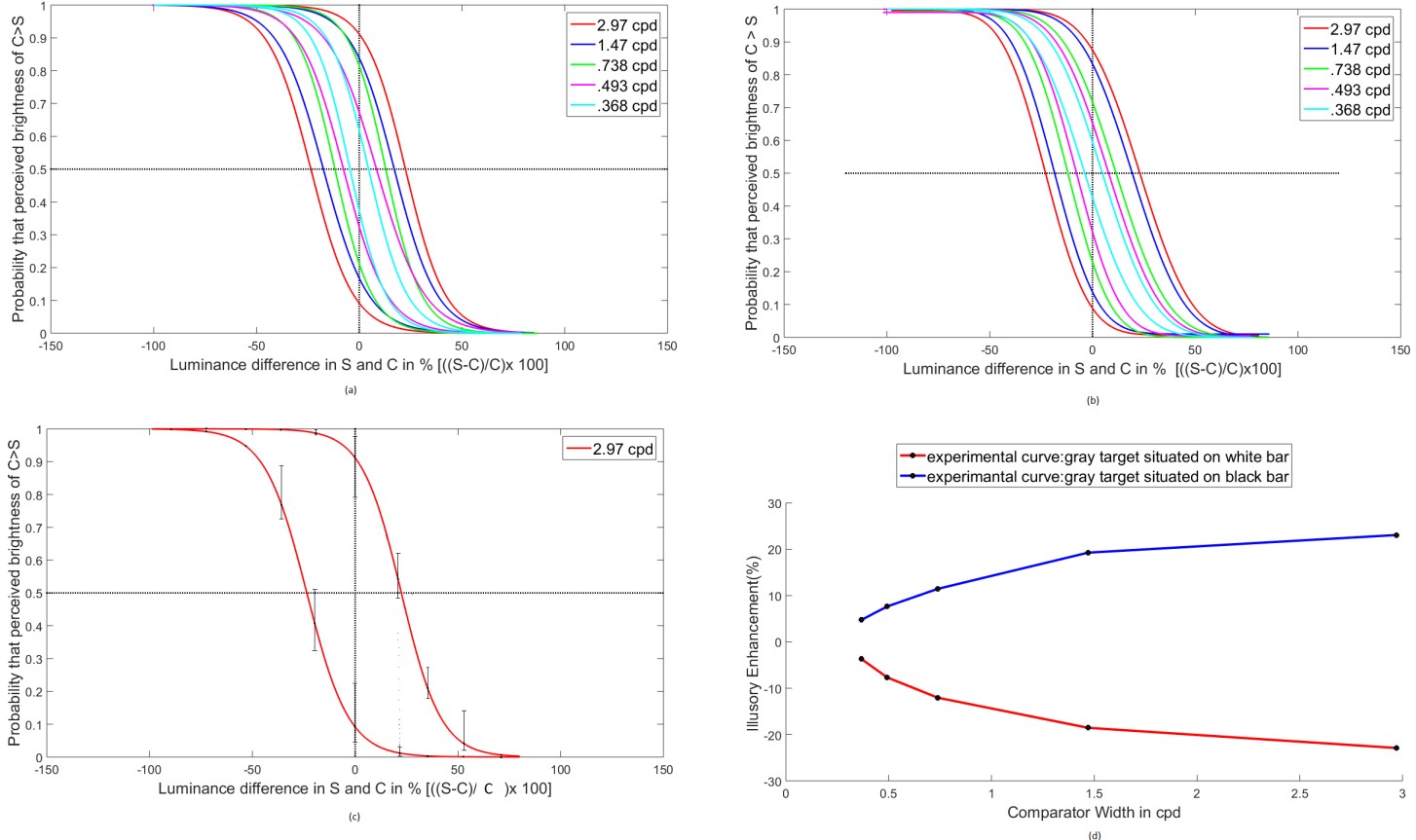

**Figure 4** **Psychophysical experimental result: average psychometric functions for different spatial frequencies are displayed in different colors.** For a particular spatial frequency, the upper curve represents the condition when the comparator appears brighter and the lower curve represents the condition when the comparator appears darker. While drawing the psychometric function, a pair of curves is placed symmetrically against the luminance difference value of 0. (A) gives the curves fitted with FitPsycheCurveLogit function. (B) gives the curves fitted with modified function by *Wichmann & Hill (2001)*. (C) gives the average psychometric function for six experimenters at a spatial frequency 2.97 cpd. The maximum and minimum deviation of perceived brightness for the individual experimenters are represented by the error bar at different luminance difference. (D) gives the perceived enhancement in percentage of the points of subjective equality for different stimulus width.

up to four input parameters, namely, the luminance difference between standard and comparator, perceived lightness of comparator, the weights for each points and targets. Abscissa in Fig. 4A represents luminance difference between standard and comparator in percentage. If the absolute gray values of the standard and the comparator are S and C, we take (S-C)/C and then multiply by 100 to get the values. The ordinate on the other hand gives the frequency or probability of observing comparator intensity to be greater than standard intensity. The line corresponding to frequency 0.5 indicates the points of subjective equality (PSE).

We have also fitted the same set of data with a modified function, developed by *Wichmann & Hill (2001)*. They presented a cumulative Gaussian function with four parameters for fitting a psychometric function. These are mean, standard deviation, guess rate (g) and lapse rate (l). The parameters g and l constrain the limits of the cumulative distribution

that provides the sigmoid shape for the psychometric curve. The plot of the same set of average psychometric data is shown in Fig. 4B. It is observed that the psychometric curves remain almost unaffected by this modification, though the family of curves appears to be more compact.

In Fig. 4C average Psychometric function for six experimenters has been plotted at spatial frequency 2.97 cpd. Observed deviations of perceived brightness for the experimenters are represented by the error bars.

In Fig. 4D, the variation of the percentage illusory enhancement with spatial frequencies is plotted. The illusory enhancement or decrement is calculated from Fig. 4B at the point of subjective equality (PSE). For example from Fig. 4B, the illusory enhancement at PSE for frequency of 0.368 cpd is less than 5%, whereas that for 2.97 cpd is about 30%. Illusory enhancement or decrement of lightness decreases as the width of the comparator is increased. The result is qualitatively similar to that obtained in *Anstis (2005)*.

## Variation with aspect ratio of the comparators

Psychometric curves, given in Fig. 5 are obtained by fitting the experimental data with the function FitPsycheCurveLogit. In the first set of experiment, as shown in Fig. 5A, the comparator width was kept fixed at eight pixels or 0.738 cpd, whereas the lengths of the comparators were varied over the set (16, 8, 4, 2) pixels or (2.76, 1.38, .69, .345) degrees. The second set of experiment is an exact repetition of the first one, only changing the width of the comparator at four pixels or 1.47 cpd. Results are shown in Fig. 5B. One may calculate the illusory enhancement or decrement from the data on points of subjective equality (PSE) as depicted in the Figs. 5A and 5B, following the same procedure that had been adopted in drawing the Fig. 4D from Fig. 4B. It is clear from the Fig. 6 that for both the spatial frequencies, the percentage illusory enhancement or decrement of lightness is almost independent of the length of the comparator, though it varies with spatial frequency. If any set of four points are connected by piecewise straight lines, the net line becomes almost parallel to the abscissa. This property is exhibited by all the four lines in Fig. 6. Identical result was observed by *Bakshi et al. (2016)* over a much larger range of lengths. In Fig. 7, lines, parallel to the abscissa, with an ordinate value equal to the average of the set of four points, are drawn. We assume a Null Hypothesis H0: the percentage illusory enhancements (or decrements) for all lengths of the comparators at any specified spatial frequency are equal to the average value of the set. To check whether such H0 is tenable or not, we performed tests for goodness of fit for all four sets (each set containing four points) with $\chi^2$ distribution. The four *p*-values for the four sets are displayed in the Fig. 7. It is observed that all the *p*-values are greater than 0.99, which leaves little scope to reject the Null Hypothesis. We, therefore, conclude: *percentage illusory enhancement (or decrement) in White's illusion depends on the spatial frequency of the inducing grating and is independent of the length of the comparator.*

## Proposed model

To explain the observations on White's illusion, we propose a simple model of assimilation, which is equivalent to an averaging over the neighbourhood. To perform that averaging,
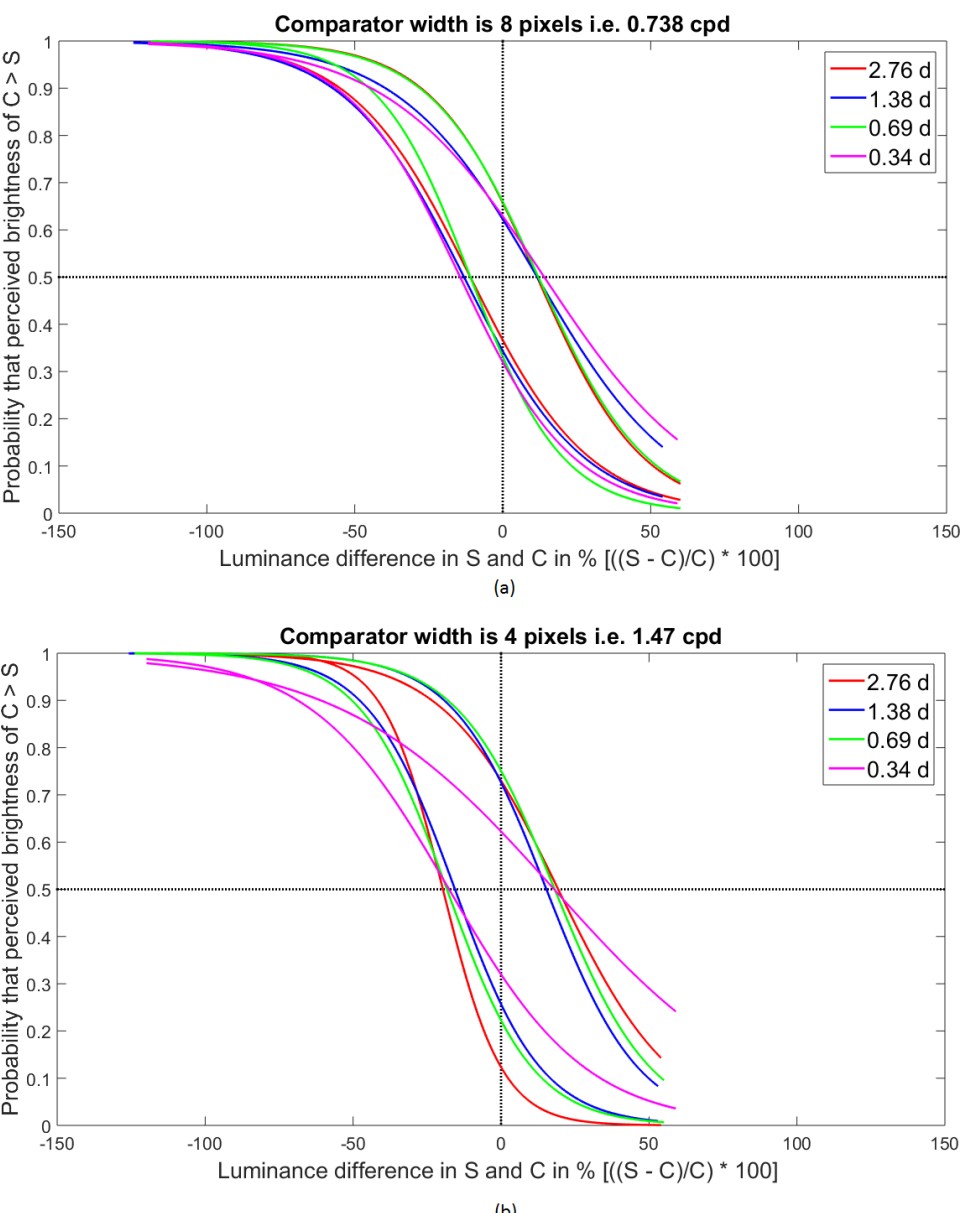

**Figure 5** **Psychophysical experimental result: average psychometric functions for the different lengths of the gray patches are displayed in different colors.** While drawing the Psychometric function, a pair of curves is placed symmetrically against the luminance difference value of 0. In (A) the width of the comparator is is eight pixels, i.e., 0.738 cpd and in (B) the width of the comparator is four pixels, i.e., 1.47 cpd.

we propose to filter the image data with a two-dimensional Gaussian $G(x,y)$ having only a single free parameter, namely, the standard deviation or scale factor $\sigma$ as given below

$$G(x,y) = \frac{1}{\sigma\sqrt{2\pi}} e^{-\frac{(x^2+y^2)}{2\sigma^2}}.$$

From the first set of experiments on White's illusion in which the spatial frequency was varied over five values, as depicted in Figures 4A or 4B, it is possible to estimate the illusory
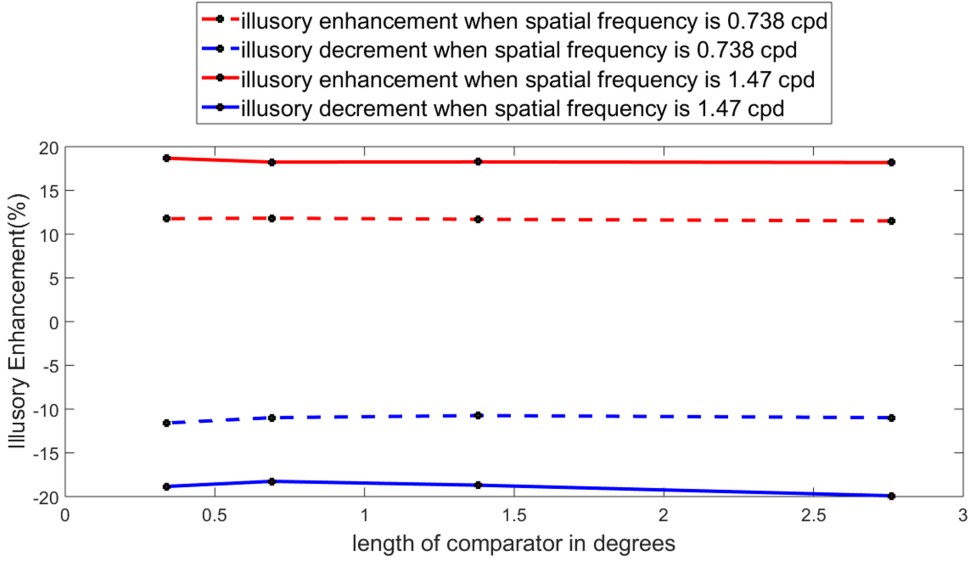

**Figure 6  Perceived enhancement for different target lengths.** Perceived enhancement in percentage of the points of subjective equality for different target lengths.

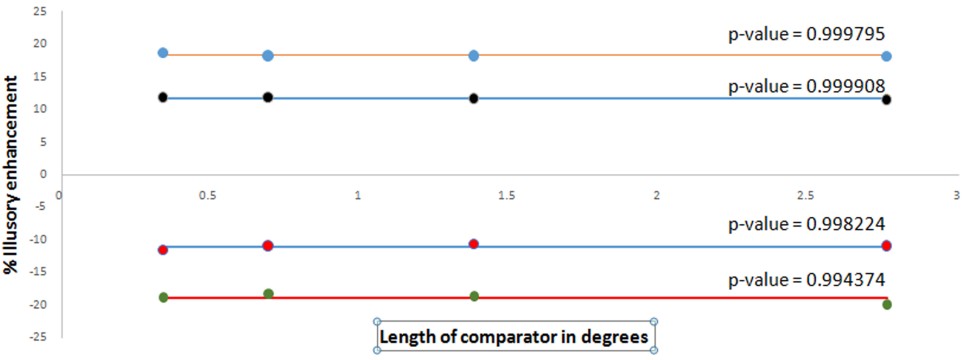

**Figure 7  Experimental data on % illusory enhancement as a function of length.** Experimental data on % illusory enhancement as a function of length of gray target in degrees has been plotted.

enhancements (decrements) at various frequencies as illustrated in Fig. 4D. To check whether the simple Gaussian averaging model, as proposed above, is adequate to simulate the experimentally obtained values of illusory enhancement, as given in Fig. 4D, one may proceed as follows. One may convolve the digital data of the image with the proposed Gaussian filter in order to measure the absolute gray value of the central point of the comparator after convolution. Since we know the actual value of the brightness/lightness of the comparator, we may easily evaluate the illusory enhancement (decrement) in percentage as depicted by our model. We call this value as the convolution response (%). By adjusting the value of $\sigma$ we bring it as close as possible to the experimentally measured values of illusory enhancement (%). Though a simple Gaussian averaging may reproduce the experimental values, the results for all the spatial frequencies, however, can not be

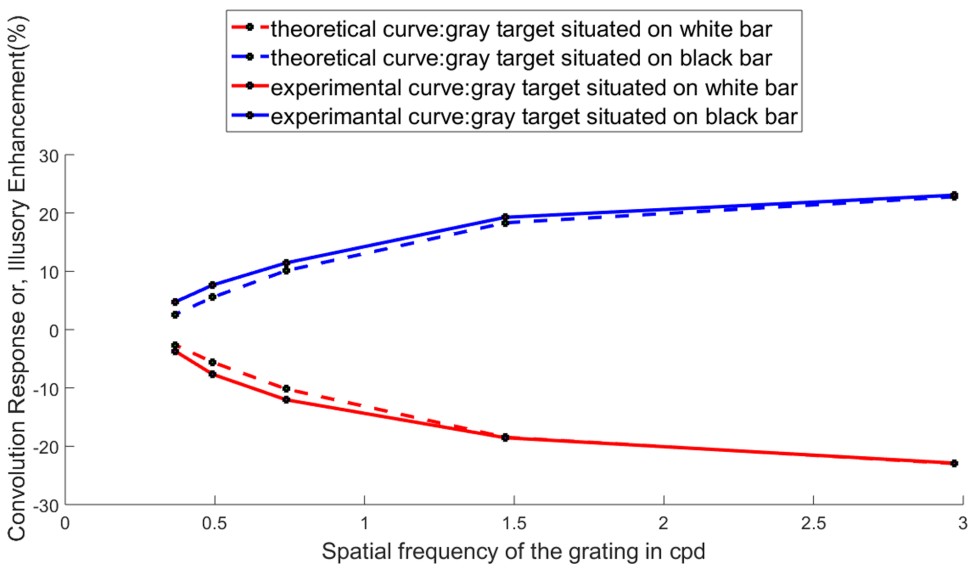

**Figure 8 Variation of the illusory enhancement (%) or the convolution response (%) with spatial frequency of the inducing grating.** The value of $\sigma$ is varied from 3.6 to 0.8 as the spatial frequency is increased. The simulated data is normalized against the intensity value of 128. The continuous curves represent the experimental results while the dotted curves are the outcome of the computer simulation.

simulated with a single value of $\sigma$. This is not surprising, because as the widths of the grating bars are increased, the corresponding mask sizes also increase. Consequently the scale factors ($\sigma$) should also increase. To get good fit with experiment, the values of $\sigma$ were chosen as (3.6, 3.0, 2.3, 1.4 and 0.8) for spatial frequencies (0.368, 0.493, 0.738, 1.47 and 2.97 cpd) respectively. The experimentally obtained illusory enhancement (or decrement) and the convolution response (%) after fitting the value of $\sigma$ are compared in Fig. 8. The length of the comparator was kept fixed in this experiment at 70 pixels or 12.07 degrees.

Coming to the second part of the experiment, namely the variation of White's illusion with the change of length of the comparator at a fixed value of spatial frequency, it may be noted that we concluded on the basis of our psychometric experiments at the end of **Results**: *Percentage illusory enhancement (or decrement) in White's illusion depends on the spatial frequency of the inducing grating and is independent of the length of the comparator.* Guided by this conclusion, a simulation experiment was conducted with five spatial frequencies (2.97, 1.47, 0.738, 0.493 and 0.368 cpd), wherein for each frequency, the length of the comparator was varied over fourteen values (70, 60, 50, 40, 32, 24, 16, 14, 12, 10, 8, 6, 4 and 2 pixels, or in other words 12.075, 10.35, 8.625, 6.9, 5.52, 4.14, 2.76, 2.415, 2.07, 1.725, 1.38, 1.035, 0.69 and 0.345 degrees respectively). For this extended study, we did not perform the psychometric experiments as each one of these experiments is quite time-consuming. Instead, we took recourse to the simulation experiment with the cue that the convolution response (%) should remain largely the same over various lengths of the comparator at any specified value of the spatial frequency. We are confident that even if the simulation experiments are replaced by psychometric experiments, the final conclusion would remain the same.

**Table 1** Parameters of the stimuli and the filter with varying spatial frequencies and the length of the gray patch.

| Length of gray patch in pixels | Scale factor | | | | |
|---|---|---|---|---|---|
| | Spatial frequency = 2.97 cpd(2 pix) Mask size = (6 × 6) | Spatial frequency = 1.47 cpd(4 pix) Mask size = (12 × 12) | Spatial frequency = 0.738 cpd(8 pix) Mask size = (20 × 20) | Spatial frequency = 0.493 cpd(12 pix) Mask size = (35 × 35) | Spatial frequency = 0.368 cpd(16 pix) Mask size = (50 × 50) |
| 70 | 0.8 | 1.4 | 2.3 | 3 | 3.6 |
| 60 | 0.8 | 1.4 | 2.3 | 3 | 3.6 |
| 50 | 0.8 | 1.4 | 2.3 | 3 | 3.6 |
| 40 | 0.8 | 1.4 | 2.3 | 3 | 3.6 |
| 32 | 0.8 | 1.4 | 2.3 | 3 | 3.6 |
| 24 | 0.8 | 1.4 | 2.3 | 3 | 3.6 |
| 16 | 0.8 | 1.4 | 2.3 | 3.1 | 5.7 |
| 14 | 0.8 | 1.4 | 2.3 | 3.6 | 7.3 |
| 12 | 0.8 | 1.4 | 2.4 | 4.9 | 8.5 |
| 10 | 0.8 | 1.4 | 2.7 | 6.15 | 9.5 |
| 8 | 0.8 | 1.41 | 3.8 | 7.2 | 10.4 |
| 6 | 0.8 | 1.53 | 5.1 | 8.1 | 11.3 |
| 4 | 0.81 | 2.35 | 6.35 | 9 | 12.1 |
| 2 | 1.32 | 3.32 | 7.8 | 9.8 | 12.9 |

The major constraint in the simulation was to adjust the scale factor $\sigma$ such that the convolution response (%) remains constant over all lengths of the comparator. Starting point of the simulation is to assume that the magnitude of the illusory enhancement (%) for a comparator of length 70 pixels (obtained through our first experiment and shown in Fig. 4D) for any particular frequency should not vary due to the variation of the length of the comparator. The starting value of the scale factor is also taken from the fitted curve as given in Fig. 8. The value of the spatial frequencies, corresponding length of the gray patch and the scale factor of the Gaussian kernels are presented in Table 1. The mask sizes were chosen approximately three times the corresponding widths of the grating bars. The simulation keeps the convolution response (%) constant over various values of the length of comparator by adapting to a suitable value of the scale of the Gaussian, as given in Table 1. This is illustrated in Fig. 9. It may be noted that the fitted scale factor of the Gaussian kernel remains constant over a fairly large range of the comparator length. It starts increasing only when the length and breadth of the comparator are comparable to one another.

Though the data obtained from our simulation experiment are given in detail in the Table 1, it may give rise to a possible misunderstanding. It may appear, as if, the convolution had been performed with a Gaussian filter only on the comparators without taking into consideration the effect of such a convolution over the entire visual presentation. To avoid any such misunderstanding, a pictorial presentation of the effect of Gaussian filter on standard, comparator and the inducing grating is included here. In Fig. 10A, an example of White's illusion at four different frequencies is given. Illusion at one frequency is separated from the other by an uniform gray background. A line AA′ has been drawn parallel to the abscissa, such that the grating bars, coaxial with the comparators, are all white. Similarly

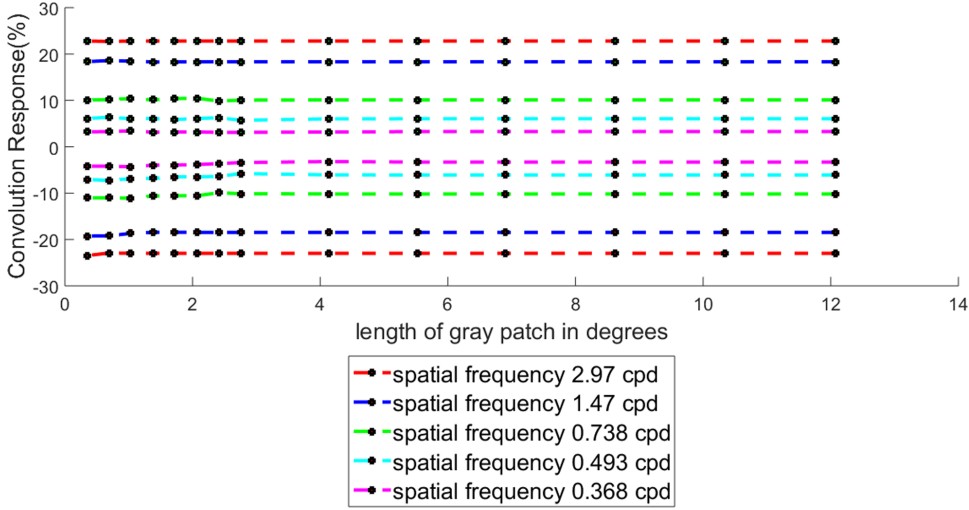

**Figure 9 Percentage convolution response for different length of the target while keeping the spatial frequency as fixed.** In the graph, the red, blue, green, cyan and magenta curves show the simulated output at 2.97 cpd, 1.47 cpd, 0.738 cpd, 0.493 cpd and 0.368 cpd, respectively.

another line BB′ has been drawn such that the grating bars, coaxial with the comparators are all black. One-dimensional intensity profiles, corresponding to the lines AA′ and BB′, are shown in Figs. 10B and 10C respectively. In both of these figures the ordinate gives the absolute gray value of the image in a scale from 0 to 255. There are four bunches of values corresponding to the four illusions considered in Fig. 10A. The absolute gray value of the uniform background that had been introduced to separate one illusion from the other is not shown in the picture. Abscissa has been labeled as "distance in pixels". This is just to provide an information about the spatial frequencies of the gratings. Distance from one bunch to the other carries no physical relevance. Figures 11A–11C show the effect of Gaussian filtering with adaptive scale factor on Figs. 10A–10C respectively. The convolved image of Fig. 10A, as given in Fig. 11A, shows blurring of both comparators and the inducing grating. However, the illusory effect on the comparators at different frequencies can be differentiated by a naked eye. Blurring effect on the inducing grating reduces as the spatial frequency is increased. The blurring effect, as shown here, is also present in the convolution output of the ODOG model by *Blakeslee & McCourt (1999)* and *Blakeslee & McCourt (2004)*. The actual intensity of the comparators in absolute gray value is 128. The Figs. 11B and 11C clearly show how the illusory enhancement through Gaussian filtering can be estimated from the diagram. For example, at a spatial frequency of 2.97 cpd, convolution output, as given in Figs. 11B and 11C, shows that the illusory enhancement in absolute gray value is (150.8–128) or 22.8, whereas the illusory decrement is (128–105) or 23. Another important feature that comes out of these figures is that the magnitude of illusory enhancement or decrement decreases as the spatial frequency of the grating is decreased.

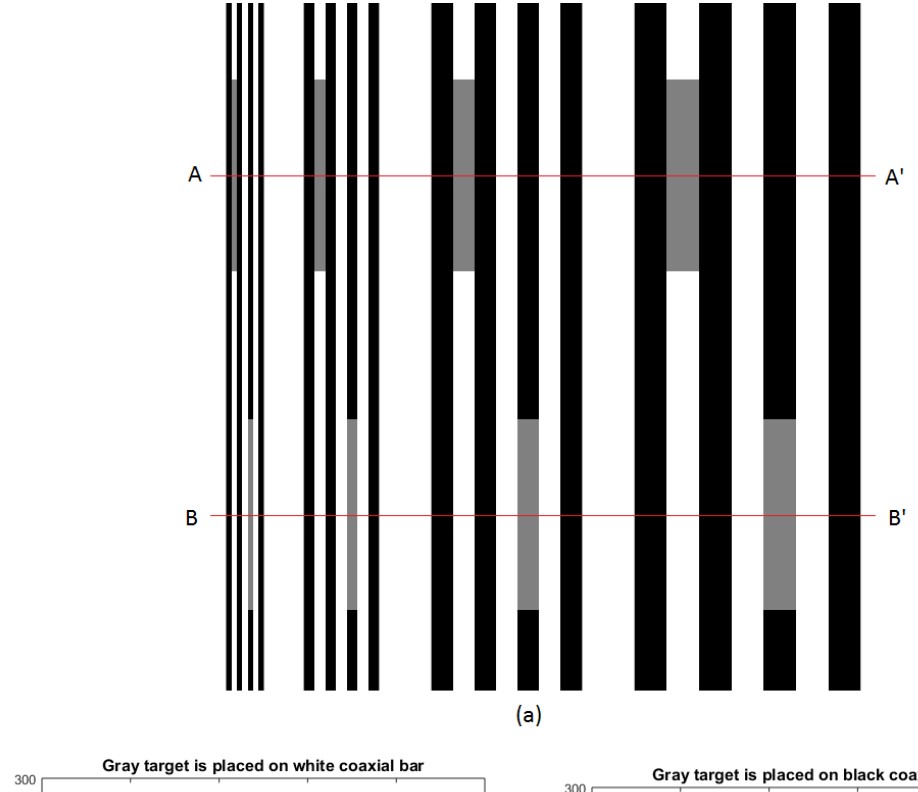

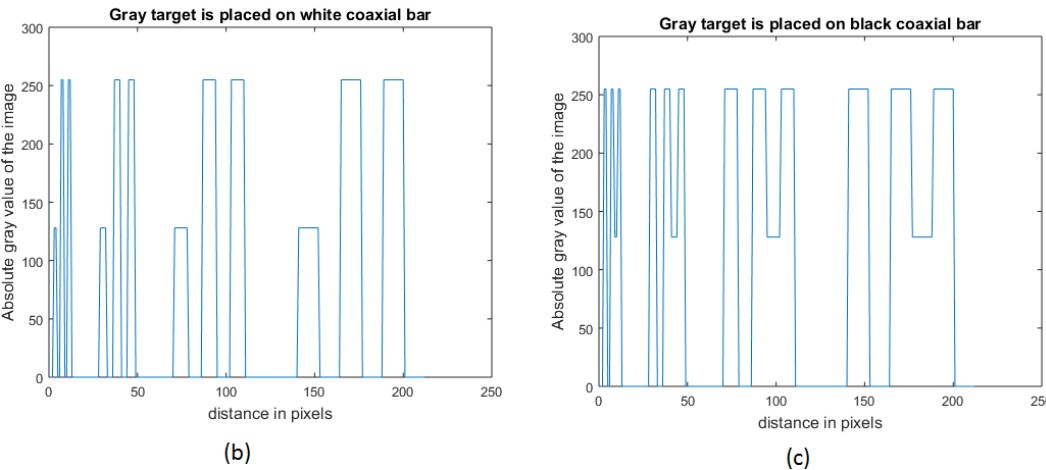

**Figure 10  White's illusion at four different frequencies separated by a uniform gray background.** (A) White's illusion at four different frequencies separated from one another by a uniform gray background. (B) and (C) represent one dimensional intensity profile corresponding to the lines AA′ and BB′, respectively.

## DISCUSSION

Regarding our proposed model, it should be noted that there is no novelty or surprise in it. It is known for a long time that a large number of cases of visual illusion may be explained fully or partially by invoking assimilation, which has a close similarity to an averaging through Gaussian kernel. We have simply emphasized here that such an extremely simple
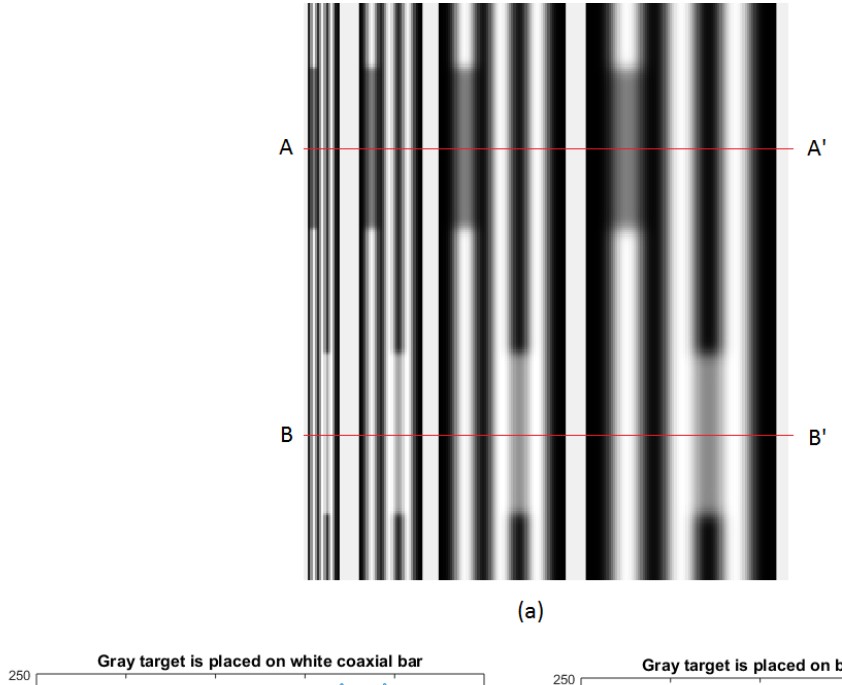

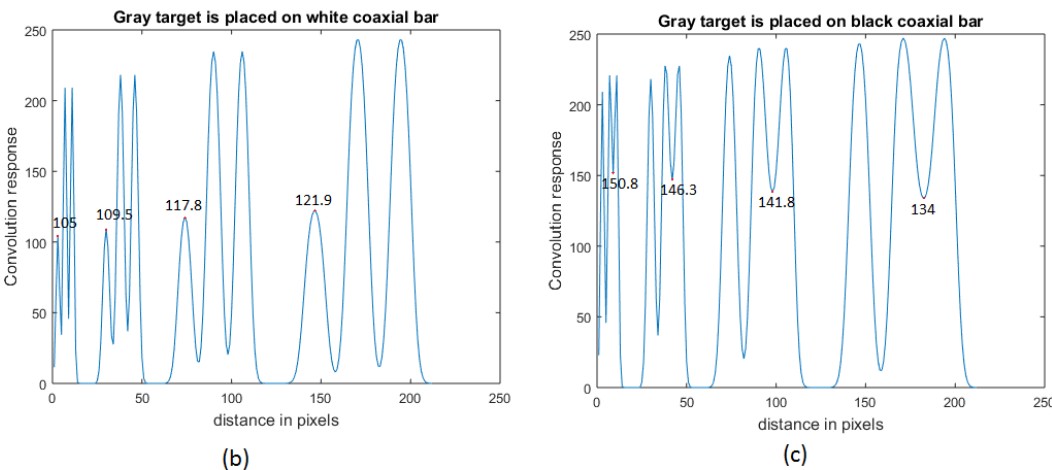

**Figure 11 The effect of Gaussian filtering with adaptive scale factors on Figs. 10A–10C.** The illusory effect at different frequencies can be differentiated by the naked eye. (A) gives convolved image of Fig. 10A with Gaussian filtering. (B) and (C) shows the effect of Gaussian convolution on Figs. 10B and 10C, respectively.

model may lead to the explanation of White's illusion over a large variation in spatial frequency and length of the comparator. We have varied the length of the comparator from 70 pixels to two pixels to show how this simple model of averaging may reproduce the observations on White's illusion over an extensive range, while the popular model involving ODOG (*Blakeslee & McCourt, 1999*; *Blakeslee & McCourt, 2004*) fails near the upper limit of this range of length (*Bakshi et al., 2016*). Moreover, the present model involves a single free parameter, namely scale factor or $\sigma$. Fitting such a wide range of data by adjusting a single parameter is undoubtedly an example of a good engineering model. To establish

this model as a paradigm in visual perception, we are to enquire further whether it is also supported by experimental neurophysiology of the visual system. It may be noted that the model involves three important features. (a) For assimilation we have used a simple Gaussian blurring. (b) The fitted scale factor varies over an wide range, indicating that in some situations information is collected and averaged over very large area. Such a long range interaction was not called for in the classical models. (c) The well-known antagonism between the central and peripheral regions in the receptive field of the neurons is given up in the present model in favour of a single Gaussian. In the following we are going to cite evidences from neurophysiology for all these three features of our model.

## Assimilation is a Gaussian averaging

Gaussian averaging is omnipresent in every layer of information collection in the retinal neuronal network. In fact in any connected network, through which the information is passed from one end of it to the other, stimulus at a point is spread through the entire network in a manner having close similarity with Gaussian averaging. A good example may be cited from the properties of resistive network in which a voltage input at any point is spread like Gaussian blurring. In the neural network of the retina itself, there are evidences of Gaussian averaging through bipolar cells, through horizontal cells and through amacrine cells.

## Variation of the scale factor over a large range

Studies on contrast started much earlier than the studies on assimilation. Experimental data and theoretical models on contrast indicate the existence of two different Gaussian averaging, one for the central and the other for the peripheral region of the receptive field. Experiments supported the belief that contrast is manifested as a local property of the network, which means the averaging are done over small areas. Consequently the value of $\sigma$ should be small. In our proposed model, we find that the fitted values of $\sigma$ vary widely, ranging from 0.8 to 13. Classical works suggest that the averaging is done over a center–surround radius of less than 300 micrometers. As a basis to our model, the experimental data from neurophysiology is necessary to support the claim of averaging over a distance, much larger than 300 micrometers. Such data are pouring over in the recent years. Data from several laboratories indicate that (*Yeonan-Kim & Bertalmio, 2016*) there are some varieties of amacrine cells which collect information over a radius of more than 500 micrometers. Recent works on wiry amacrine cell reveal that (*Manookin et al., 2015*) soma of some of these cells collects information from a distance, which may be larger than 1,000 micrometers or one mm. Diameter of such a receptive field may exceed two mm, which is larger than the classical estimate at least by an order of magnitude. These wide-field cells are also found to lack the property of center–surround antagonism. Receptive fields for these wiry amacrine cells were mapped and they were at least 10 times larger than the receptive fields of parasol ganglion cells. So the wide range of fitted value of $\sigma$ should not be considered as a mere fanciful hypothetical extrapolation. On the contrary the range corroborates to the recent data on neurophysiology.

Even if the existence of such wide field cells are accepted, one may wonder whether depending on the nature of the stimulus, different types of cells are invoked for averaging

in different types of scenario. Alternatively, is it true that the same cell, depending on the visual presentation, goes on changing its value of $\sigma$? The experimental neurophysiology of the visual pathway is still in its infancy to provide a clear answer to these queries. We shall, however, revert back to this point little later.

### Well-known antagonism between central and peripheral region vanishes in the proposed model. What is the neural correlate of that phenomenon?

Contrast sensitivity is so ubiquitous in sensory physiology, that it was predicted much before any advent in electrophysiology of neurons. Later in the very initial stage of experimental neurophysiology, the principle of lateral inhibition (existence of DOG filter) in explaining contrast sensitivity was so firmly established in visual perception that nowadays it is accepted without any controversy. Examples of assimilation started gathering much later. Though the model of lateral inhibition is at least one step more complicated than the model of assimilation, acceptance of the latter is not spontaneous. Even today, DOG is a more natural choice than the averaging through a single Gaussian. Some people believe that the existence of DOG filter is supported by direct neurophysiological evidences. That belief, however, is not so well founded. Experiments vindicating the process of DOG are all based on the stimulus response relationship of the neurons in the visual pathway. For example, an electrode is placed inside, say, a ganglion cell. A strong spot of light is shown in the centre of its receptive field to elicit vigorous response in the ganglion cell. The response becomes milder as the radius of the spot of light is increased, showing the average response from the central region is antagonistic to the average response from the peripheral region. However, nobody has been able to prove that the neurons of peripheral region release inhibitory neurotransmitter to its presynaptic terminal. Two neurons communicate either through an electrical synapse or a chemical synapse or through some ephaptic coupling. In case of retinal lateral inhibition, no signature of inhibitory neurotransmission has ever been detected. We quote from *Kramer & Davenport (2015)*: ''Despite decades of research, the feedback signal from horizontal cells to photoreceptors that generates lateral inhibition remains uncertain. GABA, protons, or an ephaptic mechanism have all been suggested as the primary mediator of feedback. However, the complexity of the reciprocal cone to horizontal cell synapse has left the identity of the feedback signal an unsolved mystery.'' On the other hand from a different set of experimental results, *Shapley et al. (1990)* concluded that ''responsiveness of the visual system to contrast is not a result of center–surround interaction, or, in other words, of lateral inhibition (as in the standard textbook accounts........). Rather, the key to understanding dependence of response on contrast is to realize that contrast dependence is primarily a result of the automatic gain control that produces light adaptation.''

In the light of the above discussion, we are trying to understand the phenomenon of visual perception with a fresh outlook, instead of getting biased with any existing model. Only experimental observation in the retinal network that can never be refuted is the Gaussian averaging. Averaging in the retinal network operates at different scales. It is done within a very short distance (central region, probably through bipolar cells),
medium distance (peripheral region, probably through horizontal cells) or large distance (as mentioned in this paper, probably through amacrine cells). This is the first stage for information collection. In the next stage a primal sketch (*Marr, 1982*) of the visual scenario is constructed by following some rules of combination of these averages. The averages may combine positively or antagonistically, linearly or non-linearly following a single principle, namely the principle of survival of the individual in the process of evolution. Obviously the rules of combination should depend on the nature of the stimulus. In some situation central and peripheral averages may combine antagonistically. In some situation the antagonism may vanish and/or may be superseded by another average from a much larger area (known as extra classical receptive field). The locus of coding these rules is still elusive. It may be at a higher cortical level or it may even be controlled by the continuous feedback and feed forward exchanges between the primary receptors and a neuron at a higher level in the visual pathway, say at lateral geniculate nucleus. To get a non-controversial neural correlates for the models of visual perception, we shall have to wait till experimental neurophysiology provides answer to these questions. Until then, we shall have to go through the process of careful data collection and prediction of empirical models. Reverting back to the discussion that we had postponed at the end of point (b) above, it appears that may be both of the alternatives are true. Existence of specialised cells which may sum up information from a long distance are indeed present, but not always called for. On the other hand the same neuron may dynamically adjust its scale factor for the survival of the organism. No fixed rules are followed while drawing even the primal sketch from the raw data. Visual perception is complicated by the fact that the raw visual information dictates the algorithm through which it should be processed for final computation.

Finally, we would like to discuss another tempting alternative model for visual perception, namely deconstruction of the visible image into its Fourier components by neurons in the circuit of visual pathway and later reconstructing the image at the cortical level from the amplitudes of the components and their phases (*Campbell & Robson, 1968*). The first and foremost requirement of such a model is to prove the existence of neurons which are finely tuned to various spatial frequencies. Such cells were discovered long back in the primary visual cortex of the cat and macaque monkeys (*Maffei & Fiorentini, 1973 ;De Valois, Albrecht & Thorell, 1982*). So it is expected that the Fourier amplitudes of the visual image may be well represented among the cells in the visual cortex. It was also found that simple and complex cells of cat encode the phase information of the image (*Spitzer & Hochstein, 1985*). If both amplitude and phase information are retractable in the cortex, reconstruction of the original visual scenario should not be problematic.

We have digressed to this topic of Fourier decomposition, because it carries some relevance to the present topic. In a previous work (*Mazumdar et al., 2016*) we had studied the variation of the width of Mach bands with the nature of discontinuity in the intensity. We defined a term ''sharpness of discontinuity'' (SOD) and observed that an empirical DOG model, in which the scale factor of the surround Gaussian is a function of SOD, is able to reproduce the results. It was further shown in that paper that if the visual system is indeed capable of performing Fourier analysis, then SOD can be estimated. The model

showed that the effect of surround suppression had to be reduced as the contrast at the edge increased. In the extreme limit of binary edges, where the contrast is maximum and represented by a step edge, no lateral inhibition takes place. Thus at high frequency, DOG gets converted into a Gaussian kernel leading to the vanishing of Mach band. Taking the cue from this previous work, we have applied that in this paper too. Since White's illusion involves many high frequency features, we conjectured that a model with total surround suppression with an adaptive scale factor may be appropriate for analysing the effect. It is, of course, merely a conjecture that has resulted into a good working model. There is no direct experimental observation supporting this conjecture.

## CONCLUSION

For the bipedal apes, walking upright, the visual system is no doubt an extremely important sensor to gather information of the outside world. From any visual scenario, the organism for its survival has to identify its predators and prey. Depending on its survival strategy, the same scenario may be analysed for its details or may be grossly averaged out. Accordingly, the final computation of the same scenario may differ widely from one another. Human beings, evolving over millions of years, have learned that the information from the same scenario may be computed through varieties of algorithms. However, though the computational algorithms may differ from one type of analysis to the other, the basic neuronal circuits in collecting the raw visual information probably do not change. Information collected at different layers are manipulated via different rules depending on the purpose of the analysis.

Proceeding with that belief, we feel that there is no necessity of having different neuronal circuits for visualising contrast or assimilation. Depending on the scenario, the same set of data may give rise to the perception of contrast or of assimilation. Such an idea was floated long back by *Helson (1963)* while working with a set of white bars on a gray background. He observed that by changing the spatial frequency of the bars, the perception moves from contrast to assimilation. A similar effect was observed by replacing the white bars with black bars. Another famous experiment by *Reid Jr & Shapley (1988)* with disk and ring stimuli showed that decreasing the size of the surrounding area decreased contrast and increased assimilation. However, these authors were so obsessed by the limited size of the center–surround receptive field that they had to assume the process of assimilation as a post-retinal, perhaps cortical, phenomenon. With the recent findings on wide field amacrine cells, that obsession is removed. Thus we agree with *Yeonan-Kim & Bertalmio (2016)* that the existence of such wide field interneurons establishes that contrast sensitivity and lightness assimilation share the same neural locus.

## ACKNOWLEDGEMENTS

The authors would like to thank Ms. Gargi Bag, CDAC, Kolkata for helping in the development of the MATLAB code. The authors are also grateful to all the reviewers for their valuable comments that helped in improving the work.

### Funding

This study was funded by MeitY, Govt. of India (project titled 'Development of Human perception inspired algorithms for solving deeper problem in Image Processing and Computer vision', Administrative approval No: DIT/R&D/CDAC/2(7)/2010 dated March 28, 2011) and MOSPI (through TAC-DCSW, CCSD, ISI), and the Cognitive Science Research Initiative, Govt. of India. The funders had no role in study design, data collection and analysis, decision to publish, or preparation of the manuscript.

### Grant Disclosures

The following grant information was disclosed by the authors:
MeitY and MOSPI (through TAC-DCSW, CCSD, ISI).
Cognitive Science Research Initiative, Govt. of India.

### Competing Interests

The authors declare there are no competing interests.

### Author Contributions

- Soma Mitra conceived and designed the experiments, performed the experiments, analyzed the data, contributed reagents/materials/analysis tools, prepared figures and/or tables, authored or reviewed drafts of the paper.
- Debasis Mazumdar conceived and designed the experiments, performed the experiments, analyzed the data, contributed reagents/materials/analysis tools, authored or reviewed drafts of the paper.
- Kuntal Ghosh and Kamales Bhaumik conceived and designed the experiments, analyzed the data, authored or reviewed drafts of the paper, approved the final draft.

### Human Ethics

The following information was supplied relating to ethical approvals (i.e., approving body and any reference numbers):

The experiments have been approved by Scientific and Technical Advisory Committee (STAC), C-DAC.

### Data Availability

The raw data are provided in the Supplemental Files.

### Supplemental Information

Supplemental information for this article can be found online at http://dx.doi.org/10.7717/peerj.5626#supplemental-information.

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
