# Peer review of "An adaptive scale Gaussian filter to explain White’s illusion from the viewpoint of lightness assimilation for a large range of variation in spatial frequency of the grating and aspect ratio of the targets"

_PeerJ, doi:10.7717/peerj.5626_

## Round 0.1 · original submission · Major Revisions

Please provide a point-by-point reply that adresses the comments from two reviewers, paying particular attention to the requests for clarification and to improve readibility. Reviewer 1's concerns about the validity of the findings should be addressed fully.

·

Basic reporting

In general the article is not particularly well written. There are places particularly in the introduction and methods where things are inconsistent and confusing. With some effort the writing of the manuscript could be greatly improved making it much more readable.

For example, in the introduction there are too many unnecessary abbreviations. I recommend cutting 'Human Visual System' HVS altogether.

Here is a truly remarkable sentence from the introduction:

" This they conclude because the White’s illusion occurs irrespective of the aspect ratio of the test patches in Figure 1, which means that although it so occurs that with respect to the two flanking bars, the lightness of the column on which the test patch is located becomes dominating beyond a certain aspect ratio as is evident from Figure 3 and Figure 4, still the direction of lightness induction does not change in such a situation."


The reference to de Weert and Spillmann misstates 'the pincushion illusion'. The use of the term 'pincushion' in their paper its to describe the shape of the target field. The Pincushion Illusion is one of illusory contours unrelated to the effect reported in their paper.

There is great inconsistency throughout the manuscripts in referring to the components of the White's Illusion stimulus. I recommend: Background (uniform grey field present across the display), Inducing Grating, and Grey Target.

A description of the phenomenology that the authors seek to explain and how/why existing models fail to account for it is kind of given in the introduction, but it is disorganized and confusing. With some work this can be made more systematic,

Spatial Frequencies are reported in cpd but target size is reported in pixels.

Luminance values should be reported, not % or RGB levels.

Ambiguity in number of bars used to define the inducing grating: text indicates a minimum of 5, the figure appears to only show 3. Clearly the authors are counting the white bars as well.

Explanation of red-indicator bars being at either top or bottom of display comes too late.

Unclear whether or not red-indicator bars would interfere in Experiment 2 with the 2/4 pixel targets. How much of the target is visible in the presence of the indicator?

2AFC: typically the stimulus that is held constant across trials would be called the standard and the one that varies the comparator. Here the gray target is held constant. Whereas the patches in the stripe are varied.

If the experiment was conducted as described then Lines 316-321 are incorrect.

Line 336 is redundant.

If methods and parameters are spelled out more clearly, table 1 is unnecessary.

Experiment 2 should show examples of the stimuli as was done for Experiment 1.

Experimental design

Aside from clarity in description, the experiments seem fine. I recommend showing at least one set of actual subject data with error bars indicating the variability observed in the results. As is, only fit-curves are shown.

The application of models appears fine; however, this sort of modeling is not really my domain of expertise and I defer to other reviewers whether the selection of parameters and how they are applied is truly appropriate.

Validity of the findings

My primary concern has to do with implicit assumption that the effects of the Gaussian Model apply only to the Gray Target without also applying to the Inducing Grating. I can simulate the Gaussian model to some degree by blurring my eyes while looking at the various figures. Indeed, doing so results in one light target and one dark one. The issue is that they appear 'solid'. That is to say that in addition to the grey target appearing either lighter or darker , the inducing grating appears darker/lighter as well. Presumably, this would be apparent if the authors included a figure similar to Figure7 showing the output of the Gaussian Model.

I note that to me I sometimes do get a filling-in of the gray target such that it does look like a solid (i.e. in front of and occluding the inducing grating). Which would be along the lines of what I would expect from the Gaussian Model. That said, the solid appears 'crisp' with sharp edges, whereas I would expect to experience some blurring with the Gaussian Model. Importantly though, I do not always experience the stimulus this way and frequently experience White's Illusion while the inducing gratings seem just as black or white as when distal to the Target.

I also note that the deWeert and Spillmann papers reveal asymmetries between induced lightening versus induced darkening. This is something that should be explored/addressed in the current paper. Perhaps their example is not a good one to use in the introduction as it may be distinct from the White's Illusion.

Additional comments

Maybe it is just me but when blurring my eye just a little bit, I note that when viewing Figure 4 the upper targets appear to get progressively lighter whereas the lower targets appear to get progressively darker to the point where for rightmost (smallest) targets, the top one looks lighter than the dark one.

Reviewer 2 ·

Basic reporting

no comment

Experimental design

no comment

Validity of the findings

no comment

Additional comments

This paper reports two experiments and a model of White’s illusion (WI), which involves perception of gray targets embedded rectangular gratings. The results showed that the strength of the illusion (different appearance of targets belonging to black and white portions of the grating) increases with increasing spatial frequency of the gratings (and thus the width of the targets), but that it does not change much with the change of the length of the targets. The model involves filters (simulated neural units) with gaussian receptive fields.
This paper is similar to a paper from the same team that I reviewed last year but is improved and involves a number of changes, including a new experiment. Furthermore, it is interesting to learn how a particular neural model reacts to WI stimuli. Some changes seem to reflect and address, in part, some of my comments. However, a few of my original concerns apply to the new paper as well.
1. The model is based on neural units with purely excitatory receptive fields, with gaussian fall-off. However, generally receptive fields of neurons in the visual system have both excitatory and inhibitory portions. Mitra et al. speculate that inhibitory portions could be suppressed by high frequency displays, such as WI stimuli, and cite some indirectly relevant literature. However, I am not aware of physiological evidence directly supporting the actual existence of such neurons, or the crucial role of high frequency stimuli in generating their receptive fields. This is relevant for assessing the import of the model, and also in light of the fact that Mitra et al. criticize other authors (Blakeslee & McCourt) on the grounds that their model involves a feature which has not been physiologically confirmed (p. 24).
2. The simulations of results for different stimuli involved gaussian filters with different parameters. Inspection of Table 2 indicates that the size of one parameter increased with spatial frequency of the gratings and decreased with the length of the target, for a total of about 30 different numerical values used in the simulations, varying approximately within a 1:16 range. No principled reason for the particular choices of these sizes for particular stimuli was offered, so that the criterion was probably goodness of fit. This makes engineering sense, but I wonder about the physiological plausibility of such a scheme. One possibility is that neurons with gaussian receptive fields with all these parameter values do exist in the visual system. If so, the question is why would, for a given stimulus, only a small subset of them which share a particular value of a parameter be activated, and none of the others? Another possibility is that the same neurons can change their receptive field parameters for different stimuli. Then the question is whether such changes could be as drastic as in the proposed scheme.
3. The authors call their model ‘multiscale’, since it uses units with receptive fields with different scale parameters. However, for any particular stimulus, they use units with a single scale to model the results, but different for different stimuli. Usually, the term ‘multiscale’ is reserved for models in which units of several different scales are all applied to all stimuli, such as in the Blakeslee&McCourt model. Such genuine multiscale models have advantages over the authors’ model, since they don’t have the problems described in the previous paragraph.

---

## Round 0.2 · accepted · Accept

Congratulations on your study!

# ·

Basic reporting

The manuscript is greatly improved!

Experimental design

No Comment

Validity of the findings

No Comment

Additional comments

Thank you for the revisions you have made to the manuscript.

Reviewer 2 ·

Basic reporting

no comment

Experimental design

no comment

Validity of the findings

no comment

Additional comments

The authors have generally taken into account my comments and incorporated them into the revised version. I have no further comments.